# Honey Bee Colony Losses in Mexico’s Semi-Arid High Plateau for the Winters 2016–2017 to 2021–2022

**DOI:** 10.3390/insects14050453

**Published:** 2023-05-11

**Authors:** Carlos Aurelio Medina-Flores, Marco López-Carlos, Octavio Carrillo-Muro, Alison Gray

**Affiliations:** 1Unidad Académica de Medicina Veterinaria y Zootecnia, Universidad Autónoma de Zacatecas, Zacatecas 98500, Mexico; lopcarmarco@uaz.edu.mx (M.L.-C.); octavio_cm@uaz.edu.mx (O.C.-M.); 2Department of Mathematics and Statistics, University of Strathclyde, Glasgow G1 1XH, UK; a.j.gray@strath.ac.uk

**Keywords:** *Apis mellifera*, COLOSS survey, colony losses, risk factors, winter mortality, Mexico

## Abstract

**Simple Summary:**

Honey bees (*Apis mellifera*) are important pollinators that benefit environmental conservation and agricultural production. Additionally, beekeeping is an important economic activity that currently faces the problem of high colony loss rates, particularly during winter. Finding solutions involves knowing the factors associated with high loss rates. To investigate whether loss rates are related to hive migration, beekeeping operation size, and *Varroa* control, we surveyed beekeepers from five states in Mexico after six consecutive winters (2016–2017 to 2021–2022). The results show an average colony loss rate of 22%, but these range from 14.9% to 30% between the years. The migration practice and beekeeping operation size did not influence the losses; however, *Varroa* control reduced colony losses. The main causes were problems related to the queen and dead colonies or empty hives, which showed differences between the winters analyzed. The results reveal high loss rates in the studied region, as has been reported in other countries. To improve the honey bee colony loss rates, it is suggested to improve the quality of queens, the control of varroasis and other diseases, and the degree of Africanization in order to reduce absconding.

**Abstract:**

The objective of this study was to estimate the loss rates of honey bee (*Apis mellifera*) colonies in six consecutive winters (2016–2017 to 2021–2022) in five states of the semi-arid high plateau region of Mexico, as well as study the factors associated with these losses. The survey included data from 544 beekeepers and 75,341 colonies. The colony loss rate differs significantly (*p* < 0.0001) between the winters analyzed and fluctuates between 14.9% and 30%, with an average of 22%. Colony losses were unaffected (*p* > 0.05) by migratory beekeeping practice and operation size, but losses were significantly affected by *Varroa* monitoring and control (*p* ≤ 0.001). The types of loss differ among the winters analyzed. In the winters of 2016–2017 to 2018–2019, beekeepers attributed higher loss rates to unresolved problems related to the queen (e.g., a queenless colony, poor laying), but in the winters of 2019–2020 to 2021–2022, the highest loss rate was the result of problems such as diseases, poisoning, and absconding (leading to dead colonies or empty hives). The results reveal high loss rates in the region studied, as reported by beekeepers from other countries. It is suggested that strategies should be implemented to improve the quality of queens, the control of varroasis and other diseases, and the degree of Africanization.

## 1. Introduction

Beekeeping is an important economic, social, and ecological activity worldwide. In Mexico, 43,000 beekeepers manage 2.2 million colonies of honey bees (*Apis mellifera*). Its significant production volumes (62,000 tons) and exports (26,000 tons) of honey place the country in ninth place and thirteenth place worldwide, respectively [1].

The importance of bees lies in the fact that they depend on the pollen and nectar of plants for their food, acting as pollinating agents for more than 75% of plant species, which favors agricultural production, conservation, and restoration of the environment [2]. It is estimated that the value of pollination in Mexico is more than MXN 63 billion (USD 1,134,000 million) [3].

One of the main characteristics of Mexican beekeeping is Africanization, a phenomenon derived from the interbreeding of the European honey bee subspecies (e.g., *Apis mellifera ligustica*) and the African subspecies *Apis mellifera scutellata* imported into Brazil in 1956 [4]. As a result of the arrival and dispersal of the Africanized bee in the country, defensive, swarming, and absconding behaviors increased. This has resulted in significant changes in the management of these bees, deaths of people and animals, abandonment of beekeeping in some cases and regions, increased production costs, and a reduction in the number of colonies and honey production relative to managing European bees [5].

Additionally, beekeeping in Mexico and the rest of the world faces various problems, among which the high levels of colony loss stands out. This is particularly important in the winter in Mexico and many other countries, when low temperatures, little or no food available in the field, the low population of bees, and the absence or reduction in numbers of drones that limit the possibilities of fertilization, cause this time of the year to be critical for the survival of honey bee colonies [6]. In the winter of 2006–2007, the annual rate of colony loss considered normal by US beekeepers was 15–20% [7], but over the past seventeen years, winter losses have generally exceeded 20% in many ecological regions, including sites where winter is not as intense [8,9,10,11,12,13,14,15,16,17,18,19,20,21]. This situation has caused beekeepers to reproduce their colonies to a greater extent in order to recover from their losses and meet production and pollination commitments, which implies considerable investments [14,22].

The high colony loss rates differ between years and regions and are caused by the interaction of multiple factors such as the *Varroa destructor* mite, deformed wing virus, *Nosema ceranae*, nutritional deficiencies, poor queen quality, poor weather conditions, pesticides, stress caused by migration, and overcrowding of hives and other practices that are related to the level of technology and beekeeping operation size [9,23,24,25]. The combined effect of these factors can inhibit the immune response of the bees, making them more susceptible to diseases and resulting in colony collapse and loss [26].

The multi-annual estimation of the colony loss rate allows us to understand the variability of losses in the studied regions over time and to identify their possible causes, thus contributing to improving the development of beekeeping [15]. The objective of this study was to determine the effect of beekeeping operation size, migration, and *Varroa* monitoring and control on the rate of loss of colonies in the semi-arid high plateau region of Mexico during six consecutive winters (2016–2017 to 2021–2022).

## 2. Materials and Methods

### 2.1. Survey Methodology and Participants

During the period from March 2017 to July 2022, a total of 657 beekeepers contacted mainly through beekeeping organizations from the states of Aguascalientes, Chihuahua, Durango, San Luis Potosí, and Zacatecas were surveyed by personal interview or electronic means to estimate the loss rate of honey bee colonies from the semi-arid high plateau of Mexico during six consecutive winters from 2016 to 2017 to 2021 to 2022. Winter was defined as the period between the moment that the beekeeper finished the pre-winter preparations for their colonies and the start of the new foraging season, which includes the period of time between the months of November and February. Only 544 respondents (whose responses concerning colony losses were consistent and error-free) were considered for data analysis (44 in 2016–2017, 104 in 2017–2018, 120 in 2018–2019, 83 in 2019–2020, 94 in 2020–2021, and 99 for 2021–2022). The surveys were applied to the same beekeeping organizations and individually sent electronically to the same addresses; therefore, it is likely that a high percentage of respondents participated consecutively in the six surveys. On average, 6.3% of the colony population in the studied region was included in the analysis every year. The sample size should allow the results to be representative of the trends of colony losses in this region of Mexico. 

The survey applied to these beekeepers was part of the annual survey carried out by the International Association for Prevention of honey bee COlony LOSSes (COLOSS), which has established standardized monitoring of winter losses of colonies in more than 30 countries simultaneously [27]. All beekeepers participated anonymously to protect their identity and encourage participation and were fully informed of the nature and purpose of the survey. The survey questions aimed to obtain information about the number of colonies going into winter and how many of these colonies were (a) alive after winter but had unsolvable queen problems (e.g., becoming queenless, poor laying, laying workers, or a drone-egg laying queen); (b) lost over winter to a natural disaster (including floods, droughts, frosts, fires, and theft); and (c) dead or were reduced to just a few hundred bees (dead colony or empty hive), caused by absconding, diseases, poisoning, or starvation, for example. In addition, beekeepers were asked if they practiced migratory beekeeping and if they monitored and/or controlled *Varroa* (*V. destructor*). In Mexico, a short version of the COLOSS questionnaire is used, including mandatory questions, so the data collected did not include, for example, information on supplementary feeding.

### 2.2. Statistical Analysis

The total number of colonies lost per beekeeper was obtained by adding the stated numbers of colonies lost to the three types of loss above, i.e., unresolvable queen problems, natural disasters, or dead colonies/empty hives. Total colony loss rates were calculated by dividing the total number of colonies lost by the total number of colonies at risk for each winter, and the results were expressed as a percentage loss rate. The 95% confidence interval (95% CI) for the loss rate was calculated using a linear model [27] for the groups of beekeepers responding each year and for the subgroups of beekeepers compared below. The frequency distribution of beekeepers with different ranges of colony loss was obtained. The percentage of beekeepers who practice migratory beekeeping was also calculated. To determine possible statistical differences regarding the rate of colony loss between beekeeping operations of different sizes (number of colonies), a classification was made according to what was described by VanEngelsdorp et al. [10] into hobby beekeepers (managing 1 to 50 colonies), secondary activity (51 to 500 colonies), and professional beekeepers (more than 500 colonies). A chi-squared test was used to assess any relationship between the proportion of beekeepers migrating their colonies and the categorized operation size. The comparison of the loss rates of colonies of the different types of exploitation between the six winters analyzed and between the potential causes of loss recorded (operation size, migrating colonies or not, and monitoring and treating for *Varroa* or not) was carried out using the Kruskal–Wallis test. The proportions of beekeepers monitoring their colonies for *Varroa* or not and the proportions controlling for *Varroa* or not were compared between years and operation sizes using a chi-squared test. All analyses were developed using SAS software [28].

## 3. Results

The rates of colony loss in the semi-arid high plateau were significantly different (Kruskal–Wallis *X*^2^ = 543, *p* < 0.0001) between the winters evaluated. The smallest percentage of colonies lost occurred in the winter of 2018–2019, and the highest was in the winter of 2019–2020. The overall loss rates and confidence intervals (95%) for the loss rates in the winters evaluated are shown in Table 1.

The number of colonies per beekeeper surveyed fluctuates between 3 and 2000, so a wide range of beekeeping operation sizes are included in this study. The colony loss rates reported by the participating beekeepers over all of the studied winters range from 0% to 78%; 54% of them presented losses of less than 20%, and 24% had losses of 21 to 30% (Figure 1). Experiencing a loss rate of more than 30% was relatively uncommon over these winters as a whole.

It was observed that in the winter of 2016–2017, 45% of the respondents had losses of more than 20% of their colonies, 44% lost more than 20% in the winter of 2017–2018, 24% in the winter of 2018–2019, 66% for winter 2019–2020, 56% in 2020–2021 and 49% in winter 2021–2022, corresponding to the years with the lower and higher overall colony loss rates above.

Considering all the beekeepers surveyed in the six winters studied, it was observed that 49% (267 of 544) of them practice migratory beekeeping. The loss rate for beekeepers who migrate their colonies (21.4%, 95% CI: 19.8−23.1%) was statistically similar to the rate for beekeepers who do not (21.5%, 95% CI: 19.6–23.3%) (and the rates of loss per beekeeper in these two groups were also not statistically different; Kruskal–Wallis *X*^2^ = 0.19, *p* = 0.66).

In relation to the beekeeping operation size, in the data as a whole and disregarding the year, it was observed that the rates of colony loss per beekeeper do not differ significantly (Kruskal–Wallis *X*^2^ = 0.37, *p* = 0.82) between hobby, secondary, and professional activities (Table 2).

The chi-squared test showed that there was a significant relationship between the proportion of beekeepers migrating their colonies and the operation size (*X*^2^ = 83.8, *p* < 0.0001); only 29% of hobby-type beekeepers migrate their colonies, while a higher proportion of secondary-activity-type beekeepers (63%) and professionals (93%) carry out this practice. We also compared the rates of colony loss for beekeepers migrating and not migrating their colonies in each operation size category of beekeepers. The results found were similar to the overall analysis. There were no significant differences (Kruskal–Wallis *X*^2^ = 0.066, *p* = 0.79) between the loss rates for hobby beekeepers who practice migratory beekeeping (*n* = 73, loss rate: 21.6%, 95% CI: 18.0–25.1%) and those who do not practice it (*n* = 178, loss rate: 21.5%, 95% CI: 19.2–23.9%), or for beekeepers in the secondary activity size category (Kruskal–Wallis *X*^2^ = 0.074, *p* = 0.78; *n* = 167 migrating, with loss rate: 21.1%, 95% CI: 19.1–23.1%; and *n* = 97 not migrating, loss rate: 21.0%, 95% CI: 18.1–23.9%). For professional beekeepers who migrate their colonies, the loss rate was 22.9% (*n* = 27), and for professional beekeepers who did not do so, it was 36.4% (*n* = 2), but due to the low sample size in this group, the statistical analysis could not be performed.

Regarding the categories of colony loss recorded in the survey, in the first three winters (2016–2017 to 2018–2019), the data from the surveyed beekeepers indicated a higher rate of loss from problems related to the queen (e.g., the colony becoming queenless, or poor laying), and for the last three winters (2019–2020 to 2021–2022) the main type of loss was the category called dead colonies or empty hives (Table 3). The rates of loss from each type of loss varied significantly between the years, from 4.6% to 9.8% for losses due to queen problems, from 2.9% to 7.2% for losses from natural disasters, and from 4.6% to 13.0% for losses characterized as empty hives/dead colonies.

It was observed that professional beekeepers had a significantly higher rate of loss due to problems derived from the queen and empty hives/dead colonies than hobby and secondary activity beekeepers (Table 4).

Finally, 75% of beekeepers surveyed indicated that they monitor the level of *Varroa* infestation in their colonies, and 84% indicated that they apply some control method for this parasitosis. There was no significant difference in either the proportions monitoring or the proportions using a control method between the winters analyzed (*p* > 0.05). No significant differences (*p* > 0.05) were found in the proportions of hobby, secondary activity, and professional beekeepers who monitor or control the *Varroa* mite. There were significant effects of monitoring and of treatment against *Varroa* on the rate of colony loss. Beekeepers who monitor the infestation level of *Varroa* had a loss rate of 20.2% (95% CI: 18.9–21.4%), while beekeepers who do not monitor lost 27.7% of their colonies (95% CI: 24.1–31.4%) (Kruskal–Wallis *X*^2^ = 14.65, *p* = 0.0001). Beekeepers who used some form of control method against the *Varroa* mite had a loss rate of 20.5% (95% CI: 19.3–21.8%), and beekeepers who did not treat had a colony loss rate of 27.7% (95% CI: 23.5–31.9%) (Kruskal–Wallis *X*^2^ = 10.58, *p* = 0.001).

## 4. Discussion

The results of the present study show that the rate of colony loss in the six winters analyzed was high (≈22%) and variable between winters, particularly between the winters of 2018–2019 and 2019–2020. This could be attributed to the different climatic conditions between years. However, other factors such as the location of the apiaries and the access of the bees to pesticides, the management practices of the beekeeper, and the genetic quality of the queens could be interacting and causing high rates of colony loss [29]. The average of the overall loss rates in the six winters analyzed exceeds the annual rate considered acceptable by surveyed beekeepers in the US (≤20%) [7,10], even without considering the losses in other seasons of the year. Additionally, 46% of beekeepers in the semi-arid highlands of Mexico had loss rates greater than 20% in the whole dataset for all years.

The loss rates of the present study are lower than those estimated in the highland and northern regions (33.4%) in the winter of 2015–2016 [17] and those of the central-western region of Mexico in the winters of 2016–2017 and 2018–2019 (24% and 25%, respectively) [30]. Across the country, the loss rates in Mexico have been 25%, 19%, 20%, and 28% in the winters of 2016–2017 to 2019–2020, respectively [18,21,31,32]. These loss rates are equally fluctuating between years and regions in other countries. In the US, winter losses ranged from 23% to 51% during the winters of 2006–2007 to 2019–2020, observing a downward trend during the first seven years after the monitoring of colony loss began (2006) and an increase in the winters of 2017–2018 to 2019–2020 [7,8,9,10,11,12,13,14,15,33].

The core project “monitoring honey bee colony losses” of COLOSS [34] has reported annual winter colony losses in more than 30 European countries, Mexico, New Zealand, and some Middle East, Asian, and African countries. They observed fluctuations between countries ranging from 2% in Bulgaria to 44% in Germany. The overall percentages of colonies lost were reported to be from 12% to 21% in the winters of 2015–2016 to 2020–2021 [18,20,31,32].

In Latin America, there are few records on the rates of colony loss. In Uruguay, a loss of 20% of colonies was reported in the winter of 2013–2014 [16]. For the winter of 2015–2016, losses ranged from 5.9% in Chile to 13.5% in Argentina, and 19% in Uruguay, while in Colombia, 10.8% of colonies lost were reported for the winters of 2014 to 2016 [19] and 50% in Brazil during the winters of 2013 to 2017 [35]. This evidence suggests that relatively high and fluctuating loss rates prevail between years and countries and generally exceed the loss rate considered “normal” by beekeepers.

It has been reported that the stress generated by the practice of migratory beekeeping makes bees more susceptible to disease and collapse [15]. However, in the present study, there was no difference in the loss of colonies between beekeepers who practice migratory beekeeping or not. Our results coincide with what was observed in beekeepers from the US and Uruguay [10,16,22]. This suggests that in Mexico and elsewhere, migratory beekeeping is not a significant factor contributing to high colony losses in winter, indicating that the stress caused by the migration of colonies is minimal or can be remedied by beekeeper management. On the other hand, Gray et al. [31] report that European beekeepers who migrate their colonies had fewer losses than beekeepers who do not. This could be the result of such beekeepers who practice migration being more experienced and/or managing more colonies.

Regarding the effect of the beekeeping operation size, it is probable that professional beekeepers have more intense management and greater colony crowding, which facilitates the transmission of diseases but better control of *Varroa*. While hobby-type beekeepers have fewer colonies, they tend to be stationary beekeepers, and their colonies are less rigorously managed. Beekeepers for whom beekeeping is a secondary economic activity tend to fall between the two groups described above [36,37]. In addition, it has been suggested that smaller beekeeping operations suffer significantly higher losses than larger operations [20,31,32]. However, some other previous studies [10,16,38] did not report differences between the different types of beekeeping operations, in agreement with the results of the present study.

The nature of the colony losses indicated by the surveyed beekeepers differs significantly between years and between the beekeeping operation sizes. For the winters of 2016–2017 to 2018–2019, the main types of lost colonies were due to losses associated with queen problems, but for the winters from 2019 to 2020 to 2021 to 2022, dead colonies or empty hives was the category with the highest loss rate, arising from problems such as diseases, absconding, and intoxications. The category of natural disasters was the one with the lowest loss rate in all winters.

Gray et al. [21,31,32] have consistently reported that in the winters of 2017–2018 to 2019–2020, the main type of loss is the category of dead colonies or empty hives. Surveyed beekeepers in the US for nine consecutive years attribute their colony losses mainly to poor queen quality, varroasis, starvation, fall colony weakness, pesticide effects, poor wintering conditions, and colony collapse disorder [7,8,10,11,12,13,14,15,39]. In Europe, inadequate treatment against *V. destructor*, American foulbrood, nosemosis, bee access to certain crops, age, and queen problems in summer have significantly affected winter colony mortality [40,41], while in Brazil, pesticides are considered the main cause of colony loss [35,42]. The observed differences in the types or causes of loss are probably due to differences in the genotype of the bees, management, and environmental conditions [43].

In addition to what has been previously mentioned about the effect of *Varroa* on the rate of colony loss in other parts of the world, Mexican beekeepers in the study region who monitored and used some control method against *Varroa* had lower loss rates than beekeepers who did not, which coincides with what was previously reported in the US where it was found that the use of varroacides was consistently associated with the lowest winter losses, particularly with amitraz [37]. In Luxemburg, low colony losses were associated with a combination of well-timed summer and winter treatments against *Varroa* [44]. Likewise, in Canada, it was found that 85% of colony deaths during winter were associated with this mite [23].

Although this study has involved a sizeable group of beekeepers in total, many of whom participated in the annual survey over multiple years, the sample size varied considerably from year to year, and this study is focused on the semi-arid high plateau region of Mexico. Wider participation of Mexican beekeepers and their organizations is required in the standardized annual surveys to generate valuable information in different agroecological regions of Mexico.

Moreover, controlled studies are needed to evaluate the effects of hive transport and distance traveled, bee genotypes, colony overcrowding levels, and management strategies on colony loss rates in order to determine specific prevention and control strategies.

## 5. Conclusions

It is concluded that the rate of colony losses in the semi-arid highlands of Mexico was moderately high (>22%) but variable within the winters analyzed. Migration of colonies and operation size did not significantly influence colony loss rates; however, monitoring and control of *Varroa* mites did have significant effects on loss rates.

In order to reduce loss rates, it is advisable to genetically improve queen quality, reduce the emergence of laying workers or a drone-egg laying queen, reduce absconding (controlling Africanization by replacing with European queens), avoid pesticides (by coordinating with farmers or improving apiary location) and improve diagnosis and control of varroasis and other diseases and pests.

At the national level, it could be important to promote relevant educational and training material for beekeepers in order to increase their knowledge and awareness of the causes of colony loss and strategies to reduce or avoid it.

## Figures and Tables

**Figure 1 insects-14-00453-f001:**
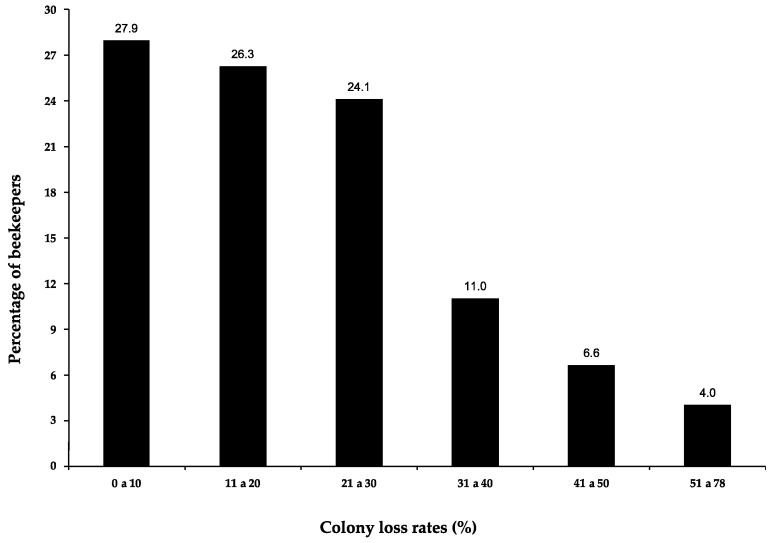
Percentage of beekeepers who presented different ranges of colony loss in the winters of 2016–2017 to 2021–2022.

**Table 1 insects-14-00453-t001:** Number of beekeepers surveyed, number of colonies before the winters, overall winter loss rate, and confidence interval (95%) for the loss rate in the winters of 2016–2017 to 2021–2022.

Winter	Surveyed Beekeepers	Number of Colonies Before Each Winter	Overall Winter % Loss Rate (95% CI)
2016–2017	44	7928	20.2 (15.2–25.2)
2017–2018	104	17,341	20.4 (17.3–23.5)
2018–2019	120	9305	14.9 (13.0–16.8)
2019–2020	83	10,966	30.0 (25.8–34.1)
2020–2021	94	12,872	23.2 (20.8–25.6)
2021–2022	99	16,929	22.3 (19.8–24.7)

**Table 2 insects-14-00453-t002:** Type of operation, number of surveyed beekeepers, number of colonies before and after winter of 2016–2017 to 2021–2022, percentage loss of the honey bee colonies, and confidence interval.

Type of Operation	Surveyed Beekeepers	Total No. of Colonies before Winters of 2016–2017 to 2021–2022	Total (%) Winter Loss Rate	Confidence Interval (95%)
Hobby	251	6984	21.6	19.6–23.5
Secondary activity	264	43,865	21.1	19.5–22.7
Professional	29	24,492	23.8	17.6–30.0

**Table 3 insects-14-00453-t003:** Rate (%) of loss (95% CI) of honey bee colonies due to problems with the queen, natural disasters, or empty hives/dead colonies during the winter seasons of 2016–2017 to 2021–2022.

	Type of Loss (%, 95% CI)		
Winter	Queen Problems	Natural Disasters	Empty or Dead
2016–2017	8.4 (5.4–11.5)	5.6 (3.1–8.2)	6.1 (3.8–8.3)
2017–2018	8.2 (5.8–10.5)	5.2 (3.5–6.8)	7.0 (5.3–8.7)
2018–2019	7.3 (6.4–8.2)	2.9 (2.0–3.8)	4.6 (3.6–5.6)
2019–2020	9.8 (7.8–11.9)	7.1 (4.5–9.6)	13.0 (10.3–15.7)
2020–2021	5.1 (3.9–6.2)	7.2 (5.3–9.1)	10.1 (8.6–13.2)
2021–2022	4.6 (3.6–5.5)	6.3 (4.5–8.2)	11.3 (9.0–13.6)
Kruskal–Wallis *X*^2^, *p*-value	28.7, 0.0001	13.3, 0.01	53.1, 0.0001

**Table 4 insects-14-00453-t004:** Rate (%) of loss (95% CI) of honey bee colonies due to problems with the queen, natural disasters, or empty hives/dead colonies during the winter seasons of 2016–2017 to 2021–2022 in different types of beekeeping operations.

	Type of Loss (%, 95% CI)		
Type of Operation	Queen Problems	Natural Disasters	Empty or Dead
Hobby	7.5 (6.5–8.5)	5.7 (4.5–6.9)	8.3 (7.0–9.6)
Secondary activity	6.4 (5.4–7.3)	5.6 (4.6–6.7)	9.0 (7.8–10.2)
Professional	9.7 (5.6–13.8)	3.1 (1.5–4.7)	10.9 (6.8–14.9)
Kruskal–Wallis *X*^2^*, p*-value	6.6, 0.03	3.1, 0.21	6.0, 0.04

## Data Availability

All data are available in this paper.

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
