# Peer review of "Honey Bee Colony Losses in Mexico’s Semi-Arid High Plateau for the Winters 2016–2017 to 2021–2022"

_insects, 2023, doi:10.3390/insects14050453_

Round 1

Reviewer 1 Report

In this study the Authors describe results from surveys of beekeepers from a semi-arid plateau in Mexico over six consecutive years regarding their overwinter honey bee colony losses and their potential causes.  The Authors used colony monitoring protocols issued by COLOSS.

The results are generally clear and the data presented will be useful for future similar studies and to monitor changes in colony losses worldwide over time. 

I just had a few minor comments:

Line 39: 'under Africanization conditions'.  Can you be more precise or give more information regarding what this means and what the consequences are for local beekeepers?

Lines 76-78:  What months does 'winter' encompass as defined here?

Lines 94:  I was surprised that the Authors did not ask survey participants about supplemental feedings since food shortages for honey bee colonies can occur over winter periods.

Lines 135-143:  How correlated are the data of beekeeper operation size and migrating colonies?  Do only 'professional' beekeepers, with many colonies, migrate their colonies for pollination?   How big is the overlap between these different  groups?  How independent, then, are the separate analyses presented here?

Lines 204 - 213:  When in relation to winter do beekeepers from this region migrate their colonies.  Does enough time elapse between events such that beekeepers have ample time to make up for losses from migration before onset of winter?

Author Response

On behalf of my co-authors, I am submitting a new version of the article "Honey bee colony losses in Mexico's semi-arid high plateau, for the winters 2016-2017 to 2021-2022." As well as a letter where the corrections to the article are exposed point by point (please see attachment).
We are grateful to reviewer for their thoughtful comments and suggestions that allowed us to improve the manuscript.

Sincerely
Dr. Carlos Aurelio Medina Flores
Correspondence author

Reviewer 2 Report

The manuscript is well-written and gives a good overview of colony losses in specified regions of Mexico over several winters. 

Author Response

Did not make corrections to the article. We thank the reviewer for their positive comments.

Sincerely
Dr. Carlos Aurelio Medina Flores
Correspondence author

Reviewer 3 Report

The results are valuable, so the manuscript is worth of publishing. However, the writing style is not adequate (manuscript is too wordy). Some suggestions are given in Comments inserted in attached PDF 

Manuscript is too wordy, so I suggest considerable shortening by a native English speaker. 

Author Response

(The authors gave the same response as above.)
